# Patient-Centered Care for Patients with Depression or Anxiety Disorder: An Integrative Review

**DOI:** 10.3390/jpm11080776

**Published:** 2021-08-10

**Authors:** Lara Guedes de Pinho, Manuel José Lopes, Tânia Correia, Francisco Sampaio, Helena Reis do Arco, Artur Mendes, Maria do Céu Marques, César Fonseca

**Affiliations:** 1Escola Superior de Enfermagem São João de Deus, Universidade de Évora, 7000-811 Évora, Portugal; mjl@uevora.pt (M.J.L.); mcmarques@uevora.pt (M.d.C.M.); cfonseca@uevora.pt (C.F.); 2Comprehensive Health Research Centre (CHRC), 7000-811 Évora, Portugal; helenarco@ipportalegre.pt; 3Abel Salazar Institute of Biomedical Sciences, University of Porto, 4050-313 Porto, Portugal; tsp.correia@gmail.com; 4NursID: Innovation & Development in Nursing Research Group, CINTESIS—Center for Health Technology and Services Research, 4200-450 Porto, Portugal; fsampaio@ufp.edu.pt; 5Higher School of Health Fernando Pessoa, 4249-004 Porto, Portugal; 6Superior School of Health, Polytechnic Institute of Portalegre, 7300-555 Portalegre, Portugal; 7Psychiatry Department, Hospital Espírito Santo, 7000-811 Évora, Portugal; mendes.art@gmail.com

**Keywords:** anxiety, depression, patient-centered care, patient care planning, symptom assessment, patient health questionnaire, patient-centered nursing, patient-focused care

## Abstract

People have specific and unique individual and contextual characteristics, so healthcare should increasingly opt for person-centered care models. Thus, this review aimed to identify and synthesize the indicators for the care process of the person with depression and/or anxiety disorders, based on patient-centered care, going through the stages of diagnostic assessment and care planning, including intervention. An integrative literature review with research in seven scientific databases and a narrative analysis were carried out. Twenty articles were included, with indicators for diagnostic evaluation and care/intervention planning being extracted. Care planning focused on people with depression and/or anxiety disorder must be individualized, dynamic, flexible, andparticipatory. It must respond to the specific needs of the person, contemplating the identification of problems, the establishment of individual objectives, shared decision making, information and education, systematic feedback, and case management, and it should meet the patient’s preferences and satisfaction with care and involve the family and therapeutic management in care. The existence of comorbidities reinforces the importance of flexible and individualized care planning in order to respond to the specific health conditions of each person.

## 1. Introduction

In 2001, the final report of Institute of Medicine’s Quality of Health Care in America recommended that in order to improve the quality of healthcare, these should be safe, effective, person-centered, timely, efficient, and equitable [1]. Moreover, according to the same report, person-centered care can be defined as the provision of healthcare that respects and is representative of patients’ individual preferences and needs. In addition, it must ensure that patient values guide all clinical decisions [1].

In this context, it is clear that healthcare must be personalized and marked by shared decision making between patients and healthcare professionals. The concept of shared decision making implies that patients’ preferences and cultural values influence clinical decisions. However, shared decision making is demanding and can be time consuming, and it may be necessary to integrate the views of both generalists and experts. In these circumstances, the values, preferences, and needs of patients must be highly considered [2], because the patient is the central focus of the care process. Shared decision making has four key characteristics: (1) Health professionals and the patient must be involved; (2) both must share the information they have; (3) a consensus must be reached based on the patient’s treatment preferences; and (4) a consensus must be reached on the treatment to be performed [3]. Thus, shared decision making must be the central principle of person-centered care, and health professionals should seek to look at the experience of healthcare through the eyes of their patient [4].

Depressive and anxiety disorders are common and recurrent mental illnesses, affecting more than 300 million people worldwide. Due to their clinical characteristics, they affect psychosocial functions, decreasing the quality of life of those who suffer from them [5].

Taking into account the current pandemic context, the prevalence of depression and anxiety tends to increase, as several studies have indicated that the pandemic has been causing an increase in the levels of stress, anxiety, and depressive symptoms [6,7,8,9,10,11,12,13]. In this sense, it is necessary and urgent to adopt healthcare aimed at surveillance, prevention, and intervention during and after the current global pandemic crisis [14]. In addition, the high prevalence of these pathologies leads to a high social and economic burden, so it is important to implement effective treatment strategies.

A recent meta-analysis concluded that person-centered care is more effective than standard healthcare for people diagnosed with depression, improving depressive symptoms and increasing the likelihood of remission [15]. In addition, he concluded that this type of care improves health-related quality of life and self-management results and decreases hospital admissions [15]. A study on late-life depression highlighted, as priority areas of research to improve health services for this clinical condition, the focus on the individual needs of the person, through patient-centered care [16]. The same study also highlighted the importance of involving informal caregivers and alternative scenarios in the care process [16]. Studies have indicated that shared decision making improves the decision-making process and the quality of healthcare in people with depression [17,18], reducing depressive symptoms in young patients [18].

Regarding anxiety disorders, little is known about the effect of person-centered care, although some studies have indicated positive results in the case of post-traumatic stress disorder [19]. Another study indicated that the cost of anxiety disorders can be reduced with greater health education, early detection, and person-centered interventions. It adds, that person-centered care planning should encourage patients to identify their strengths, preferences, and abilities to carry out activities and to focus on areas they have control over [20].

That said, it is therefore essential that health professionals carry out a timely diagnostic assessment of the person, create an appropriate care plan adapted to their characteristics and the context in which they are inserted [21], and implement person-centered healthcare. However, we did not find in the scientific literature studies that clearly define diagnostic assessment strategies, care planning, and intervention centered on people with depression and/or anxiety disorders.

### 1.1. Objective

We aimed to identify and synthesize the indicators for the care process of the person with depression and/or anxiety disorders, based on patient-centered care, going through the stages of diagnostic assessment, care planning, and intervention.

### 1.2. Review Questions

This review was conducted to answer the following questions:

What are the patient-centered care strategies used in the assessment of a person with depression and/or anxiety disorder?

What are the patient-centered care strategies used in planning care for a person with depression?

What are the patient-centered care strategies used in the intervention for a person with depression and/or anxiety disorder?

## 2. Materials and Methods

### 2.1. Protocol and Registration

The protocol for this review was registered with PROSPERO (CRD42021235405) and then published, so more details about the methodological procedures can be found in the protocol article [22].

### 2.2. Study Design

This integrative review was reported according to Preferred Reporting Items for Systematic Reviews and Meta-Analyses (PRISMA) Protocols Statement [23,24] and the methodology followed that of reference [25].

### 2.3. Eligibility Criteria

The inclusion criteria were as follows: Studies in which participants were diagnosed with a depressive disorder and/or an anxiety disorder, regardless of the state of evolution and the presence or absence of other conditions; and studies that address person-centered care, either at the diagnosis assessment, in their care planning, or in their implementation of interventions.

The inclusion criteria for the study design were empirical primary studies of quantitative or experimental observation, qualitative studies, mixed studies, and theoretical studies.

The exclusion criteria were as follows: Studies in which participants presented with symptoms of depression or anxiety but did not have an identified medical diagnosis; studies that did not address person-centered care; and studies whose study design did not meet the defined criteria.

### 2.4. Search Strategy

In this revision, a comprehensive bibliographic search was developed, and the consulted databases were: MEDLINE (with full text), PsycINFO, Scopus, Psychology and Behavioral Sciences Collection, CINAHL Plus^®^ (with full text), Web of Science, and PubMed.

The research strategy was adapted according to each database and was restricted to the last 10 years, from 2011 to 2021, so as to obtain the most recent data. Papers in the English, Portuguese, French, German, and Italian languages were reviewed.

### 2.5. Search Terms and Boolean Operators

This research included the combination of three key concepts, according to the Medical Subject Headings (MeSH) terms: Patient-centered care, depression, and anxiety. The search phrase was: ((“Patient Care Plan*”) OR (“Patient-Centered Care”)) AND ((Depression) OR (“Depressive Disorder”) OR (Anxiety)). Initially, exploratory research was carried out without limitations. However, considering the high number of results, the search was limited to the title, abstract, and/or keywords according to each database.

### 2.6. Data Collection and Analysis

#### 2.6.1. Selection of studies

The selection of studies was developed across several stages. The resultant papers found during the search of each database were exported into Mendeley and duplicates were removed. To minimize bias, two reviewers independently assessed the inclusion of the studies by reading the titles, abstracts, and keywords, excluding those that do not fit the inclusion criteria (Flowchart 1). A third reviewer was consulted in case of disagreements or doubts. Afterward, we proceeded onto the assessment stage of the complete texts using the same method.

#### 2.6.2. Data Extraction

Data extraction was performed by the same two reviewers responsible for selecting the studies, independently, and doubts and disagreements were resolved, again, by consulting a third reviewer.

In the data extraction phase, initially, a descriptive evaluation of each study was carried out using an extraction instrument designed to extract information according to the research questions.

#### 2.6.3. Quality Appraisal

For the evaluation of quantitative, qualitative, or mixed studies, we chose to use the Mixed Methods Appraisal Tool (MMAT), as it is a tool that is limited to assessing essential criteria and can provide a more efficient assessment [26]. Once again, this step was carried out by two reviewers independently, and any disagreements with the evaluation of the quality of the studies were resolved, once again, by a third reviewer. The results of the evaluation of the quality of each study were not assessed against the inclusion/exclusion criteria, so all studies selected up to this stage were included [25]. In addition, the evaluation carried out was only qualitative, without obtaining a score, as suggested by the authors who developed the tool [26]. In this way, we familiarized ourselves with the quality of the evidence produced within the scope of this review regarding quantitative and qualitative studies, with studies of lesser quality being considered less in the narrative synthesis, but were, however, not excluded [25]. Evaluation of the included articles that were not qualitative, quantitative, or mixed studies, such as theoretical studies, was not carried out.

#### 2.6.4. Strategy for Data Synthesis

As a review that includes studies with several methodologies, the synthesis and analysis of the results were of a narrative nature, structured as to answer the questions presented regarding the investigation [25]. The tools used for the diagnostic evaluation of the person are presented, as well as the care planning strategies, including the intervention, based on the care centered on persons with depression and/or anxiety disorder.

## 3. Results

The research produced 1148 results, and, after the removal of duplicates, 684 publications were identified as eligible. Based on abstract information, 158 articles were selected for an exhaustive assessment (Figure 1). The main reasons for exclusion were articles that did not address psychiatric disorders, depression, and/or anxiety disorders (43 studies), articles that did not address patient-centered care (27 studies), and articles that did not answer the research questions in terms of diagnostic evaluation and/or care planning indicators (including interventions) (18 studies).

Twenty studies were included in the review, the majority of which were carried out in the USA (15 studies), two in Germany, one in Malaysia, one in Canada and France each, and one in the United Kingdom. Of these, 17 articles addressed depression, four of which addressed, in addition to this pathology, other concomitant nosological conditions: Depression and diabetes (one study), multiple sclerosis and depression (one study), heart disease and depression (one study), and chronic pain and depression (one study). Regarding anxiety disorders, four articles addressed this pathology, three of which addressed post-traumatic stress disorder.

Regarding the diagnostic evaluation, we organized the data found, according to what each of the studies evaluated and the tools that were used (Table 1).

Table 2 shows the results of studies organized by title, year and authors; method and sample; objective; indicators for diagnostic evaluation; and indicators for care planning and/or intervention used (Table 2).

The quality of 12 studies with quantitative methodology, four with qualitative methodology, two with mixed methods, and two theoretical approaches (attached) was evaluated.

Based on the extracted indicators and content analysis, we built the patient-centered care model depicted in Figure 2.

## 4. Discussion

In fulfilment of the initially mentioned objective, and with the intention of answering the questions asked, this review included studies that allowed us to carry out a narrative analysis about the indicators of the diagnostic evaluation and planning of care and/or intervention of people with depression or anxiety disorders, taking into account patient-centered care approaches. Most studies focused on depression, with anxiety disorders being less addressed, with emphasis on post-traumatic stress disorder.

### 4.1. Diagnostic Assessment

Self-assessment instruments have been inserted in the new models of healthcare to promote person-centered care, and must satisfy their needs [39]. In addition, a systematic review of the literature concluded that self-assessment and hetero-assessment tools for depression are complementary and have identical clinical results. Therefore, these must be used to assess health results [40].

Most of the studies used scales to assess a person’s “depressive and/or anxious symptoms, the most used being the Patient Health Questionnaire Depression Scale” (PHQ-9) [27,28,29,30,31,32]. In fact, this scale was considered by a recent study as the best tool for evaluating the results reported by patients with depression [39]. In addition, unlike other depression scales, the PHQ-9 includes nine items that are based on the Diagnostic and Statistics Manual for Mental Disorders, 4th Edition (DSM-IV) [41].

Regarding the assessment of anxious symptoms, only three studies used assessment scales [19,28,34]. A study of veterans with post-traumatic stress disorder used a specific scale for this anxiety disorder [36].

The quality of life was assessed in two studies [19,33]. Of note is the fact that one of the scales used focuses on functionality with regard to mental and physical health, assessing physical functioning, anxiety, depression, fatigue, sleep disturbance, social functioning, and pain interference [19]. Another study assessed the functional state of mental health [35]. Still, regarding functionality, a study assessed psychiatric functioning and distress using the Outcome Rating Scale (ORS), which is a self-assessment instrument with four dimensions: (a) Anguish or individual or symptomatic well-being; (b) anguish or interpersonal–relational well-being in intimate relationships; (c) anguish or social well-being at work/school or in the wider social domain; and (d) general sense of well-being [38]. The aforementioned literature review recommends that for the diagnostic evaluation of people with depression, tools should be used to assess symptoms and health-related quality of life [40].

Two conducted studies involved a very comprehensive diagnostic evaluation, evaluating beyond the symptomatology the perception of self-care abilities, the assessment of self-management of mental health, positive mental health, and social participation and the use of coping strategies [28]. Another of the studies assessed the care process and the characteristics of patients with major depression using the Depression Outcomes Module, psychiatric comorbidities, the acceptability of treatment with antidepressants, the perception of treatment for depression, and the state of mental and physical health and social support [32]. The assessment of patients’ perception of the patient-centered care process was carried out in another study with dichotomous questions such as: “Asked about your concerns and questions?,” “Told about changes you could make in your daily life that could improve depression (e.g., exercise)?,” or “Given written information about depression/treatment?” [27].

Two of the studies also carried out a family assessment using the Family Assessment Device (FAD) [34] and two instruments for assessing family functionality [37]. The latter also evaluated the psychiatric symptoms of the parents [37].

In addition to the application of scales, we emphasize the fact that the diagnostic evaluation was mostly carried out by health professionals using interviews [19,32,34,35,36].

A review of the literature concluded a scarcity of studies evaluating the functionality and side effects of medication [40], despite its importance. This was equally verified in this revision. In addition, most studies, focused on the assessment of depressive and/or anxious symptoms, while studies assessing other areas such as satisfaction with the care process, quality of life, or an evaluation are still rare, despite its importance for person-centered care.

### 4.2. Care and Intervention Planning

With regard to care planning, in the studies selected in this review, it was possible to find several recommendations, many of which converged on the need to target the plan to patients and their individual conditions. An example of this is a study carried out with people with cardiac pathology and depression, whose conclusion was that interventions at the level of mood must be flexible to respond to the unique needs of each person. In addition, care planning must be personalized, identifying the problems that contribute to depression and that patients choose to work on [42]. It also appears that considering the preferences, concerns, and needs of a patient’s daily life in care planning, in addition to contributing to positive results in the treatment of depression, is strongly related to satisfaction with the care provided [27]. Thus, it is recommended that there is a personalization of care, focusing on the patient’s daily life. Meeting patient preferences, especially with regard to self-care and treatment, represents a criterion of quality of care [29,34,35,36,37,42,43].

Still within the scope of respect for the individuality of the person, it is recommended that patients be involved as partners in the identification of care and treatment planning objectives, taking into account the problems identified by each one [28,35,36]. In this sense, a literature review concluded that psychological support techniques, such as problem solving techniques, behavioral activation, and motivational interviews, are useful in supporting the involvement of persons with depression in their care plan in order to achieve their goals [44].

There are studies that have gone further and concluded that patient participation in decision making regarding their therapeutic plan is beneficial for the development of treatments and their results, the so-called shared decision-making process [28,34,35,36,38,42,43,45].

In this sense, one of the studies aimed to develop a strategic tool for the promotion and implementation of shared decision making in the use of antidepressants by patients with major depression. Based on the opinions of patients and doctors, six main themes were identified: Summary of treatment options; correct ways of taking medication; potential side effects of medication; sharing the case study regarding treatment options; cost of treatment options; and information from the pharmacist [45]. In addition to patient involvement were recommendations for the involvement of the caregiver/family—something highly valued and identified by patients as important [29,32,35].

Another way of involving the patient in their therapeutic process was mentioned in one of the studies included in this review that studied, in patients with diabetes and depression, the use of a technological platform for self-monitoring symptoms of depression and to alert health professionals. This study concluded that the self-monitoring system for the depressive symptoms under study has the potential to make healthcare more patient-centered by improving depression monitoring and care management, even in resource-limited settings [30].

In addition to the above, patient satisfaction should be considered when planning care for people with depression, as well as the factors that influence this satisfaction [27,32,35,46]. Regular proactive follow-up and collaborative patient-centered care were considered by people with depression to be factors of satisfaction. On the contrary, the lack of empathy and the mechanization of care delivery were factors of dissatisfaction [32,35,46]. Attending to patient satisfaction during the care process is an important indicator and is strongly related to the quality of healthcare. In 77.8% of the studies, patient satisfaction was positively related to an improvement of the clinical results and patient safety [47]. The evidence also demonstrates a relationship between adherence to the therapeutic regime and satisfaction with the care provided, since patients tend to trust health professionals more when they are satisfied with the care process [48].

One of the studies also used a systematic patient feedback system that obtained positive results, with readmission rates below the national reference values [38]. Meanwhile, yet another study recommended carrying out a relapse prevention plan [29].

Still following the logic of patient care centrality, some studies included in their healthcare planning the improvement of health literacy through information and education [29,32,35,38].

Management of adherence to the therapeutic regimen should also be taken into account in the planning process [30,32,45]. It is important to know which strategies guarantee better adherence to the therapeutic regimen. Likewise, the management of the side effects of medicine must be considered in the planning of care provision, in order to ensure the best adherence to treatment [32,43,45].

In addition to the above, in some of the studies, we found that the management of care planning was carried out by a case manager [33,35,42,46]. Regular patient follow-up was also mentioned [32,33,46]. In this sense, one of the studies in the present review developed a complex intervention carried out by nurses based on case management and user preferences [42]. In fact, a literature review concluded that nurses play a key role in managing care for people with depression and other complex medical conditions, as they have a significant impact on depressive outcomes having been trained to see the patient as a whole [44]. Now, this approach is fundamental for the development of person-centered care plans [44]. Successful interventions by nurse case managers include regular patient follow-up, symptom registration, treatment monitoring, goal setting, and education [44]. However, these indicators were also found in some studies of the present review, as mentioned above.

There is also the recommendation of the provision of care to respect a model of collaboration in the community [29]. One of the studies focused on the provision of collaborative care, centered on persons with major depressive disorder, which were compared to the standard intervention. Although there were no differences between groups in reducing depressive symptoms, health professionals were perceived as more “participatory” in the therapeutic process and as being more useful in identifying the individual needs of each person and in promoting their adherence to treatment [35]. Also noteworthy is telemedicine-enhanced antidepressant management (a type of intervention in a stepped care model for people with depression), which substantially improved user satisfaction and perceptions that care was centered on their individual needs [32].

Regarding the treatment of post-traumatic stress disorder (PTSD), a study identified the need for treatment to address common problems such as anger, nightmares, sleep, depression, or relationship difficulties, and suggested that if trauma-focused psychotherapy does not resolve these, joint strategies should be looked into [36]. Two studies addressed the use of complementary and alternative medicine, such as yoga, meditation, tai chi, and mindfulness, considering them as care models centered on patients with PTSD [19,49].

Regarding person-centered pharmacological interventions, the relevance of antidepressants should be selected according to the individual needs of each person, and, for this, four clusters of symptoms should be considered: Anxiety, fatigue, insomnia, and pain [50]. Of note also is the fact that women in the perinatal period prefer non-pharmacological interventions rather than the use of antidepressants [34].

Patients tend to highlight the importance of empathic listening and empathic action as a vehicle to feel more understood, valued, and truly cared for [51]. Health professionals, on the contrary, tend to emphasize the importance of familiarity with the user, teamwork, and the flexibility/continuity of care, so that they are more centered on the person [29].

Despite the above, it is necessary for clinicians to consider the limitations of person-centered care for patients with depression. For example, the decision-making ability of a person with major depression or psychotic depression may be affected, limiting the provision of person-centered care. A review of the literature concluded that depression can impair decision-making skills, with appreciation being the most impaired skill [52]. As limitations of this review, we highlight the great heterogeneity of the extracted results, which makes narrative analysis difficult. In addition, the search was carried out by title, abstract, and/or keywords, given the exhaustive number of articles identified, with languages limited according to the domains of the researchers.

## 5. Conclusions

In short, what stands out most in the evidence found in this review is the importance of care being centered on the person, given the positive results that the studies obtained. It is important that an adequate diagnostic evaluation is carried out using the self-monitoring of symptoms and that this is part of the care planning. Thus, person-centered care planning should include the identification of a person’s specific problems; personalized care planning that responds to the individual needs of the patient and with the establishment of individual treatment goals; shared decision making; patient information and education; existence of a systematic patient feedback system with the self-monitoring of symptoms; care based on a case management model with regular monitoring of patients; meeting of the patient’s preferences and satisfaction with care; family involvement in care; elaboration of a relapse prevention plan; management of adherence to therapy; and management of the side effects of medication. Thus, care planning must be individualized, dynamic, flexible, and participatory. This is a relevant contribution to clinical practice, as it provides some data that may be conducive to more person-centered care provision.

In addition, several studies have focused on anxiety and/or depression comorbidity with other organic pathologies, which reinforces the importance of recommending flexible and individualized care planning in order to provide specific responses to each person’s health condition.

It is suggested, for future investigation, that literature reviews identical to this one be carried out, but that they address other psychiatric pathologies, as well as others focused on person-centered strategies for the promotion of mental health and disease prevention.

## Figures and Tables

**Figure 1 jpm-11-00776-f001:**
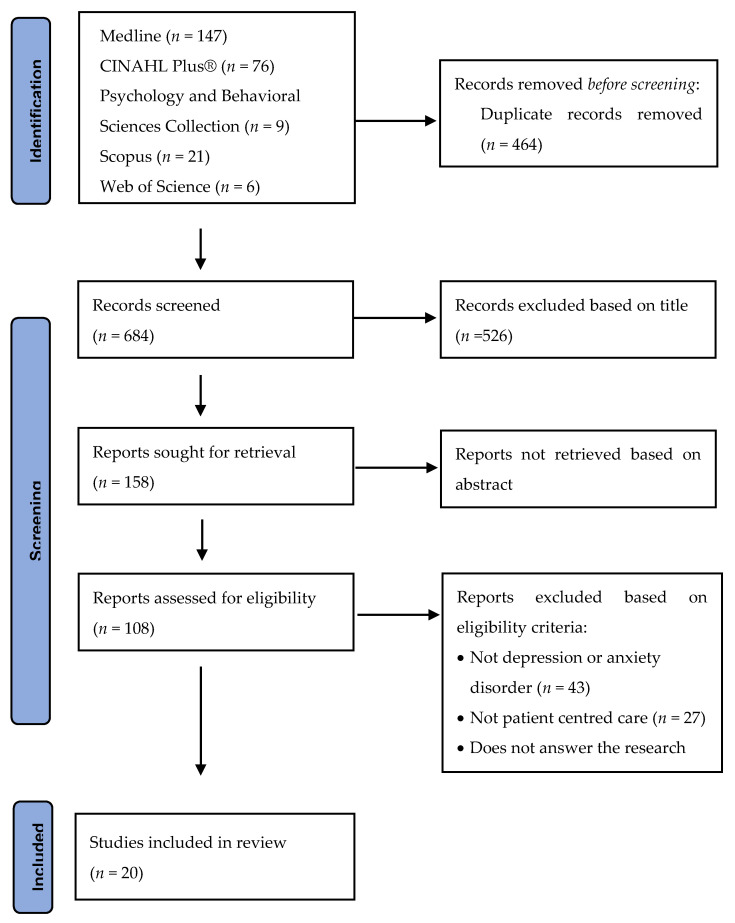
PRISMA flowchart.

**Figure 2 jpm-11-00776-f002:**
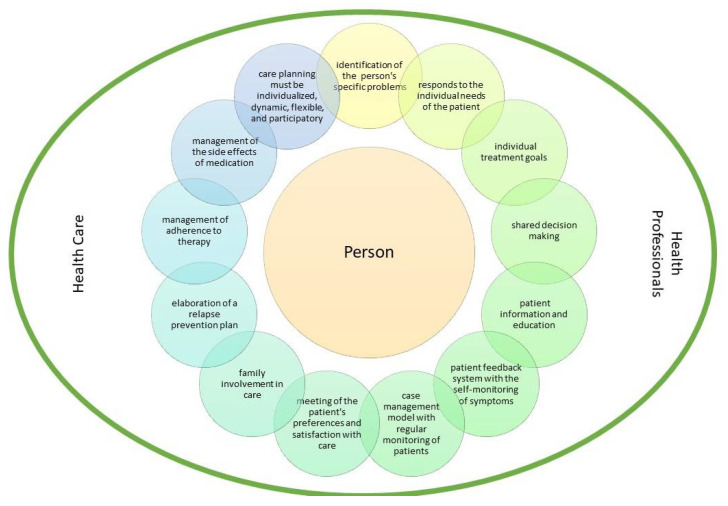
Patient-centered care model.

**Table 1 jpm-11-00776-t001:** Patient assessment indicators and assessment tools.

Indicator	Tools of Assessment	Studies
Depressive symptomology	Severity of depressive symptoms: Patient Health Questionnaire 9 (PHQ-9)	[27,28,29,30,31,32]
	Beck’s Depression Inventory (BDI-II)	[33,34]
	Center for Epidemiological Studies Depression Scale (CES-D)	[35,36]
	Hamilton Rating Scale for Depression (HAM-D)	[34]
	Composite International Diagnostic Interview (CIDI)	[35]
	Hopkins Symptom Checklist (SCL-20)	[32]
	Child Depression Rating Scale—Revised (CDRS-R)	[37]
	Children’s Depression Inventory—Youth and Parent Versions (CDI)	[37]
Psychiatric functioning and distress	Outcome Rating Scale (ORS)	[38]
Mental health functional status improvement	Medical Outcomes Study (MOS-SF12)	[35]
Anxiety/stress symptomology	Generalized Anxiety Disorder 7 (GAD-7)	[28]
	Hamilton Rating Scale for Anxiety (HAM-A) and State–Trait Anxiety	[34]
	Inventory (STAI)	[34]
	Perceived Stress Scale-4 (PSS-4)	[19]
Symptoms of post-traumatic stress disorder	PTSD Checklist (PCL)	[36]
Self-management	Mental Health Self-Management Questionnaire (MHSQ)	[28]
Degree of participants’ positive mental health	Mental Health Continuum—Short Form (MHC-SF)	[28]
Perception of self-care abilities	Therapeutic Self-Care Scale PAM-13	[28]
Health-related quality of life:	Short-Form 8 Health Survey (SF-8) and EuroQol-5D	[33]
	Patient-Reported Outcomes Measurement Information System (PROMIS) 28	[19]
Self-efficiency	General Self-Efficacy Scale (SWE)	[33]
Social participation	Social Participation Scale	[28]
Coping strategies	Brief COPE	[28]
Family assessment	Family Assessment Device (FAD) Conflict Behavior Questionnaire (CBQ) Children’s Expectations of Social Behavior Questionnaire (CESBQ)	[34]
Parent psychiatric symptomatology	Behavior Symptom Inventory (BSI)	[37]
Youth problem behaviors	Child Behavior Checklist (CBCL)	[37]
Social support	Duke Social Support and Stress Scale	[32]
Process of care and the characteristics of patients with major depressive disorder	Depression Outcomes Module	[32]
Patient-centered care process	Patient Assessment of Chronic Illness Care (PACIC), specific to depression care	[27]
Health status	Short Form (SF12V)	[32]
Perceptions about depression treatment	Depression Health Beliefs Inventory	[32]
Acceptability of antidepressant treatment	Quality Improvement for Depression	[32]
Psychiatric comorbidity	Mini International Neuro-psychiatric Interview	[32]

**Table 2 jpm-11-00776-t002:** Results of the studies and extraction of indicators.

Title/Author(s)/Year	Type of Study/Sample	Objective	Diagnostic Assessment: Indicators and Tools	Care Planning/Intervention
Patient Feedback as a Quality Improvement Strategy In An Acute Care, Inpatient Unit: An Investigation of Outcome and Readmission Rates Reese et al., 2018	Benchmarking methodology with a naturalistic data set: RCT + literature revision and comparison Sample: 2247 patients in a psychiatric care unit in the community: 51.5% mood disorders; 39.8% schizophrenia and other PP; 4.9% substance abuse disorder; 2.4% anxiety disorders; 1.3% others	To assess the effectiveness and readmission rates of services provided to racially and ethnically diverse patients, at or below the federal poverty line, in a 32-bed psychiatric center that implemented the Partners for Change Outcome Management System (PCOMS) as a strategy for quality improvement.	Psychiatric functioning and distress: Outcome Rating Scale (ORS)	Shared decision making Information and education Personalized care planning Systematic patient feedback system: Partners for Change Outcome Management System (PCOMS) A quality improvement strategy, based on patient-centered care, that uses a systematic patient feedback system. It promotes the focus being centered on involvement, benefit, and shared decision making through systematic feedback during hospital treatment. The dimensions of patient-centered care include improving health literacy through information and education, coordination and integration of care, physical comfort, emotional support, and personalized care, which encompasses the concept of shared decision making.
Effectiveness of Telephone-Based Aftercare Case Management for Adult Patients with Unipolar Depression Compared to Usual Care: A Randomized Controlled Trial Kivelitz et al., 2017	RCT Sample: 199 people with major depression or dysthymia	To assess the long-term effectiveness of post-hospital telephone follow-up, based on case management for patients with depression.	Severity of depressive symptoms: Beck’s Depression Inventory (BDI-II) Health-related quality of life: Short-Form 8 Health Survey (SF-8) and EuroQol-5D Self-efficiency: General Self-Efficacy Scale (SWE)	Case manager Establishment of the individual objectives of treatment Regular monitoring of patients
The Effects of Patient-Centered Depression Care on Patient Satisfaction and Depression Remission Rossom et al., 2016	Longitudinal, observational study Sample: 792 people diagnosed with depression from 83 primary care clinics	To explore the specific aspects of patient-centered care that are best associated with improving depression and satisfaction with care.	Severity of depressive symptoms: Patient Health Questionnaire 9 (PHQ-9) Evaluation of the patient-centered care process: Patient Assessment of Chronic Illness Care (PACIC), specific to depression care	Personalized care planning for everyday life Patient preferences Establishment of the individual objectives of treatment Shared decision making Patient satisfaction
Profiles of Recovery from Mood and Anxiety Disorders: A Person-Centered Exploration of People’s Engagement in Self-Management Coulombe et al., 2016	Quantitative transversal study Sample: 149 people with anxiety (36.9%), depressive disorders (55.7%), or bipolar (36.2%)	To identify the profiles underlying mental health recovery, describe the characteristics of the participants corresponding to each profile, and examine the associations of the profiles with criteria variables.	Severity of depressive symptoms: Patient Health Questionnaire 9 (PHQ-9) Severity of anxiety symptoms: Generalized Anxiety Disorder 7 (GAD-7) Degree of participants’ positive mental health: Mental Health Continuum–Short Form (MHC-SF) and Self-management Mental Health Self-Management Questionnaire (MHSQ) Perception of self-care abilities: Therapeutic Self-Care Scale Social participation: Social Participation Scale Coping strategies: Brief COPE	Establishment of the individual objectives of treatment: Personal Project System Rating Scale (PSRS) Shared decision making
Workforce Development to Provide Person-Centered Care Austrom et al., 2016	Quality Study 16 (ACC) Sample: 16 care coordination assistants (CCA) and 73 users aged 65 and over with at least one code of diagnosis for dementia and/or depression	To describe the development of a competent workforce committed to providing patient-centered care to people with dementia and/or depression and their caregivers; report qualitative analysis of the workforce’s case reports about their experiences; present the lessons learned on the development and implementation of a community-based care collaboration model, using a new workforce referred to as CCA.	Severity of depressive symptoms: Patient Health Questionnaire 9 (PHQ-9)	Patient preferences Family/caregiver involvement Information and education Relapse prevention plan IMPACT model (Unützer et al., 2002; http://impact-uw.org, accessed on 1 April 2021) of depression at the end of life. The team worked with the patient’s primary care physician to develop and implement a care plan for depression. If patients do not improve, they can be referred to psychiatry. The care plan includes: - Providing education on depression to the patient and caregiver; - Training the patient and the caregiver in behavioral activation and scheduling pleasant events; - Problem-solving therapy (taught by trained and licensed staff, a registered nurse (RN), or a Master of Social Work (MSW)); - Monitoring the symptoms of depression using the PHQ-9 to respond to treatment; - Completing a relapse prevention plan with each patient who has improved; - Antidepressant therapy, prescribed by the patient’s family doctor, if appropriate and necessary.
Perinatal Antidepressant Use: Understanding Women’s Preferences and Concerns Battle et al., 2013	Prospective study Sample: 61 pregnant women with (*n* = 31) and without (*n* = 30) depression in the second trimester, between 19 and 39 years of age	(1) To characterize the experiences and difficulties of women in decision making regarding treatment for depression, including attitudes related to the prenatal use of antidepressant drugs among participants who have experienced prenatal depression. (2) To evaluate hypothetical references for the treatment of postpartum depression among all participants (regardless of the state of depression).	Depressive symptoms: Hamilton Rating Scale for Depression (HAM-D) and Beck Depression Inventory (BDI) Anxiety symptoms: Hamilton Rating Scale for Anxiety (HAM-A), State–Trait Anxiety Inventory (STAI), and Family Assessment Family Assessment Device (FAD)	Shared decision making Patient preferences 71% of depressed women received pharmacological or non-pharmacological treatment during pregnancy (five psychotherapy, six selective serotonin reuptake inhibitor (SSRI), and 11 combined treatment (psychotherapy plus SSRI)).
Patient-Centered Technological Assessment and Monitoring of Depression for Low-Income Patients Wu et al., 2014	RCT Sample: 444 patients with diabetes and depression	To assess and monitor the depressive symptoms of diabetic patients in primary healthcare.	Severity of depressive symptoms: Patient Health Questionnaire 9 (PHQ-9)	Self-monitoring of symptoms Management of adherence to therapy
The Symptom Cluster-Based Approach to Individualize Patient-Centered Treatment for Major Depression Lin and Stevens, 2014	Theorical work	To present a theory and evidence for an individualized patient-centered treatment model for severe depression, designed around a cluster-based approach for the selection of antidepressants.		Antidepressant treatment strategy, structured to provide the best patient-centered care in the management of depressive disorder The choice of antidepressants should be guided by the presence of one to four groups of common symptoms: Anxiety, fatigue, insomnia, and pain—each of which can be treated effectively by an appropriate class or classes of antidepressants.
Towards Personalizing Treatment for Depression: Developing Treatment Values Markers Wittink et al., 2013	Observational quantitative study Sample: 86 people with depression	To describe and demonstrate a method for developing “value markers” or profiles, based on the relative importance of depression treatment attributes.		Patient preferences Management of the side effects of medication Shared decision making
Patients’ Perspectives on Depression Case Management in General Practice—A Qualitative Study Gensichen et al., 2012	Quality study, cluster-randomized controlled trial Sample: 41 people with depression (aged 18–80 years) that attended primary care	To explore patients’ perceptions about the practice-based case management of depression, their satisfaction with it, and the way in which living with depression contextualizes case management.		Patient satisfaction Case manager Regular monitoring of patients Effectiveness of case management for patients living with major depression Case management was provided for 12 months by health professionals, who monitored symptoms and adherence to medication through regular telephone contacts. A semi-structured interview was conducted.
The UPBEAT Depression and Coronary Heart Disease Program: Using the UK Medical Research Council Framework to Design a Nurse-Led Complex Intervention for Use in Primary Care Barley et al., 2012	Systematic review and quality study, using the guidelines of the Medical Research Council (MRC), to develop and evaluate complex interventions; iterative evidence review; focus group study (quality) to shape the intervention	To develop a nursing intervention, based on primary care, to improve mood and cardiac outcomes in patients with coronary heart disease and depression.		Flexibility and personalization of the care plan Answer the individual needs of the patient Identification of the problems (patient and health professional) Shared decision making Case manager Patient preferences Development of a complex intervention, performed by nurses in primary healthcare, based on the case manager
Relationship between Satisfaction, Patient-Centered Care, Adherence and Outcomes Among Patients in a Collaborative Care Trial for Depression Deen et al., 2011	Randomized controlled trial Sample: 360 people diagnosed with depression in primary care	To explore the relationship between satisfaction, patient-centered care, adherence to antidepressants, and clinical outcomes in a collaborative care model for depression.	Process of care and the characteristics of patients with major depressive disorder: Depression Outcomes Module Psychiatric comorbidity: Mini International Neuro-psychiatric Interview Social support: Duke Social Support and Stress Scale Acceptability of antidepressant treatment: Quality Improvement for Depression Perceptions about depression treatment: Depression Health Beliefs Inventory Health status: Short Form (SF12V) Depression severity: Hopkins Symptom Checklist (SCL-20) e Severity of depressive symptoms: Patient Health Questionnaire 9 (PHQ-9)	Family/caregiver involvement Patient satisfaction Information and education Management of adherence to therapy Management of medication side effects Regular monitoring of patients Evaluation of patient-centeredness of care using the Experience of Care and Health Outcomes Survey (ECHO) Model of staggered treatment of depression for a period up to 12 months Treatment intensity was increased for patients who did not respond to lower levels of care, involving a greater number of staff with increasing knowledge of mental health. The team included primary care providers and external units, such as nurses specializing in depression, pharmacists, and tele-psychiatrists. The specialist nurses, in caring for persons with depression, made phone calls to them to provide the following interventions: Education, assessment of barriers, monitoring of symptoms, adherence to medications, and medication side effects. They followed guidelines for addressing specific treatment barriers, reasons for non-adherence (e.g., concerns about addiction), and specific side effects. Telephone meetings between pharmacists and patients who did not respond to treatment included a history of medication and ongoing side effect management. Psychiatrists supervised the team off-site and provided consultations via interactive video.
Disentangling Multiple Sclerosis & Depression: An Adjusted Depression Screening Score for Patient-Centered Care Gunzler et al., 2015	Retrospective cohort study Sample: 3507 people diagnosed with multiple sclerosis and depression	To develop a depression assessment tool to better evaluate the depressive symptoms in people with multiple sclerosis.	Severity of depressive symptoms: Patient Health Questionnaire 9 (PHQ-9)	
Comorbid Chronic Pain and Depression: Patient Perspectives on Empathy Sternke et al., 2016	Quality study Sample: 18 patients with chronic pain and comorbid depression	To analyze patients’ perspectives of the emerging theme of empathy and to describe how patients build their experiences and expectations around empathic interactions.		Empathic interaction
Comparative Effectiveness of Standard versus Patient-Centered Collaborative Care Interventions for Depression among African Americans in Primary Care Settings: The BRIDGE Study Cooper et al., 2013	Cluster randomized trial Sample: 132 African-American patients with major depressive disorder	To compare the effectiveness of standard collaborative care interventions with patient-centered collaborative care, culturally adapted for African-American patients with major depressive disorder over 12 months of follow-up.	Depressive symptoms: Center for Epidemiological Studies Depression Scale (CES-D) and the Composite International Diagnostic Interview (CIDI) Mental health functional status improvement: Medical Outcomes Study (MOS-SF12)	Patient preferences Establishment of the individual objectives of treatment Shared decision making Family/caregiver involvement Patient satisfaction Information and education Case manager Collaborative patient-centered care: The ability of healthcare providers to look at patients as unique people, build an effective relationship, use the biopsychosocial model to explore the beliefs, to understand the values and meaning of the disease for the person, and to find a common basis with regard to care plans. Likewise, both patient centrality and cultural competence emphasize the health system’s ability to align services to meet patients’ needs and preferences. Standard collaborative care: For example, structured approaches to care based on the principles of chronic disease management and using depression care managers to work together with primary care professionals and a mental health specialist to monitor mental health and medicines, coordinate care, and facilitate patient involvement.
Development of a Strategic Tool for Shared Decision-Making in the Use of antidepressants among Patients with Major Depressive Disorder: A Focus Group Study Zaini et al., 2018	Focus group Sample: 19 doctors and 11 patients with major depression	To develop a strategic tool for the promotion and implementation of shared decisions on the use of antidepressives amongst patients with major depression.		Shared decision making Management of adherence to therapy Management of medication side effects
Parent and Youth Preferences in the Treatment of Youth Depression Langer et al., 2021	Exploratory study Sample: 64 young people and 63 parents with depression	(1) To identify variations in the preferences of parents and young people for the treatment of depression. (2) To explore the relationships between parental and youths demographics and psychosocial functioning, and the preferences that parents and youths recommend	Depressive symptoms: Child Depression Rating Scale-Revised (CDRS-R) and Children’s Depression Inventory—Youth and Parent Versions (CDI) Youth problem behaviors: Child Behavior Checklist (CBCL) Parent psychiatric symptomatology: Behavior Symptom Inventory (BSI) Family functioning: Conflict Behavior Questionnaire (CBQ) and Children’s Expectations of Social Behavior Questionnaire (CESBQ)	Patient preferences (using the Initial Treatment Preferences Questionnaire)
The Use of Yoga in Specialized VA PTSD Treatment Programs Libby et al., 2012	Exploratory study, mixed, transversal Sample: 125 coordinators of treatment programs for people with PTSD	To investigate the prevalence of the use of 32 types of complementary and alternative medicines in specialized treatment programs for PTSD.		Use of yoga, mindfulness, and meditation as therapeutic resources
Presenting Concerns of Veterans Entering Treatment for Posttraumatic Stress Disorder Rosen et al., 2013	Randomized controlled study Sample: 216 veterans with post-traumatic stress Disorder (PTSD) in outpatient treatment and 812 in residential treatment; other than PTSD, 45.8% were depressed and 19.4% suffered from anxiety disorders	To identify which problems veterans expect to improve with treatment; to analyze if there are differences between the problems presented by veterans in outpatient treatment and veterans in residential treatment, as well as between genders and how much time they served; to assess if veterans expect PTSD treatment to be more effective in solving some problems more than others.	Symptoms of Post-Traumatic Stress Disorder (PTSD) Checklist (PCL) Depressive symptoms: Center for Epidemiological Studies Depression Scale (CES-D)	Involving the patient in identifying the problem Establishment of the individual objectives of treatment Shared decision making Treatment of PTSD, moving from psychoeducation to active psychotherapy, and moving from psychotherapy centered on the present to psychotherapy centered on trauma. The results confirm the importance of educating patients about how effective treatments available can relate to personal goals. Clinicians should be prepared to offer interventions or provide referrals for common problems such as anger, nightmares, sleep, depression, or relationship difficulties, if these problems are not resolved with trauma-centered psychotherapy or if patients are unwilling to undergo this type of treatment.
Participating in Complementary and Integrative Health Approaches Is Associated with Veterans’ Patient-reported Outcomes Over Time Elwy et al., 2020	Longitudinal cohort survey Sample: 119 veterans with PTSD	To examine the results reported by veteran patients over the time they have integrated CIH (complementary and integrative health).	Quality of life/functionality: Patient-Reported Outcomes Measurement Information System (PROMIS) 28 Stress assessment: Perceived Stress Scale-4 (PSS-4) and PAM-13 (assesses the person’s knowledge and their aptitudes and trust regarding health self-management and self-care)	The intervention consisted of the involvement of veteran patients in their own self-care through non-pharmacological treatment, through body–mind practices (tai chi, meditation, and yoga), which were coordinated with traditional medicine.

## Data Availability

Not applicable.

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
