# Peer review of "Patient-Centered Care for Patients with Depression or Anxiety Disorder: An Integrative Review"

_jpm, 2021, doi:10.3390/jpm11080776_

Round 1
Reviewer 1 Report
Thank you for allowing me to review this article. I found the topic interesting, but the writing is in need of a thorough editing to update grammar and punctuation issues. The paper and Tables should not use full justification. The tables are difficult to read because of the uneven spacing between the words. In reviewing the instructions to the authors, I did not find any reference to full justification.
I also read in the author's instructions that the citations were to be numbered. I thought I was reading an APA paper with the author, year citations. There is a lack of consistency with the way the et al.' s are presented though out the paper.
While the paper was presented in a logical sequence and the selection process for choosing papers to include was excellent some of the sentences need to be rephrased as they are confusing. For example, the sentence on p. 2 "In 2016, a review study and interviews with specialists in late depression,, highlighted as priority areas of research to improve health services for this clinical condition the focus on the individual needs of the person, through patient-centered care (Hoeft et al, 2016).
Sections 2.1 and 2.2 require the reader to go to two different sources to understand the protocol and study design. The author's should not expect the reader to understand these topics without some explanation. Please consider adding a little more explanation so the reader does not have to stop reading to do research.
I feel the article is worth publishing if changes are made to improve the readability and consistency in the article.
Author Response
Dear reviewer,
Firstly, we would like to thank you very much for your analysis and for the opportunity to improve our paper. We found your recommendations extremely useful and we are sure they helped improve the overall quality of the manuscript.
We tried to address all your recommendations and to give response to all your comments.
We send the manuscript for linguistic review to improve English language. We modify the tables formatting and the citations format. We add a little more explanation in Materials and Methods section.
We hope that the changes we have made are in line with what you have suggested.
Best regards,
LP
Reviewer 2 Report
Dear Authors, Dear Editor,
I find your paper interesting and valuable. The article deals with an important issue that will most likely become an increasingly important aspect of psychiatric care. I do not have any major remarks on the article, please note only two minor comments:
- what are the limitations of person-centered care? could Authors refer for example to care of patients with severe episodes of depression with psychotic features (e.g. delusions)?
- please improve formatting of Table 2.
Kind regards,
JS
Author Response
Dear reviewer,
Firstly, we would like to thank you very much for your analysis and for the opportunity to improve our paper. We found your recommendations extremely useful and we are sure they helped improve the overall quality of the manuscript.
We tried to address your recommendations and to give response to your comments.
- We add the limitations of person-centered care in discussion.
- We improve formatting of Table 2.
We hope that the changes we have made are in line with what you have suggested.
King regards,
LP